# Non-Ergot Dopamine Agonists and the Risk of Heart Failure and Other Adverse Cardiovascular Reactions in Parkinson’s Disease

**DOI:** 10.3390/brainsci14080776

**Published:** 2024-07-31

**Authors:** James A. G. Crispo, Nawal Farhat, Yannick Fortin, Santiago Perez-Lloret, Lindsey Sikora, Rebecca L. Morgan, Mara Habash, Priyanka Gogna, Shannon E. Kelly, Jesse Elliott, Dafna E. Kohen, Lise M. Bjerre, Donald R. Mattison, Renée C. Hessian, Allison W. Willis, Daniel Krewski

**Affiliations:** 1McLaughlin Centre for Population Health Risk Assessment, University of Ottawa, Ottawa, ON K1G 5Z3, Canada; 2Department of Biostatistics, Epidemiology and Informatics, Perelman School of Medicine, University of Pennsylvania, Philadelphia, PA 19104, USA; 3Human Sciences Division, NOSM University, Sudbury, ON P3E 2C6, Canada; 4School of Epidemiology and Public Health, University of Ottawa, Ottawa, ON K1G 5Z3, Canada; 5Observatorio de Salud, Pontificia Universidad Católica Argentina, Consejo de Investigaciones Científicas y Técnicas (UCA-CONICET), Buenos Aires C1107AAZ, Argentina; 6Departamento de Fisiología, Facultad de Medicina, Universidad de Buenos Aires (UBA), Buenos Aires C1121ABG, Argentina; 7Health Sciences Library, University of Ottawa, Ottawa, ON K1H 8M5, Canada; 8Department of Health Research Methods, Evidence, and Impact, McMaster University, Hamilton, ON L8S 4L8, Canada; 9Aboriginal Cancer Control Unit, Cancer Care Ontario, Toronto, ON M5G 2L7, Canada; 10Department of Public Health Sciences, Queen’s University, Kingston, ON K7L 3N6, Canada; 11Cardiovascular Research Methods Centre, University of Ottawa Heart Institute, Ottawa, ON K1Y 4W7, Canada; 12Department of Family Medicine, University of Ottawa, Ottawa, ON K1G 5Z3, Canada; 13Institut du Savoir Montfort, Ottawa, ON K1K 0T2, Canada; 14Risk Sciences International, Ottawa, ON K1P 5J6, Canada; 15University of Ottawa Heart Institute, University of Ottawa, Ottawa, ON K1Y 4W7, Canada; 16Department of Neurology, Translational Center of Excellence for Neuroepidemiology and Neurological Outcomes Research, Perelman School of Medicine, University of Pennsylvania, Philadelphia, PA 19104, USA

**Keywords:** Parkinson’s disease, non-ergot dopamine agonists, heart failure, cardiovascular reactions, medication safety, systematic review, meta-analysis, randomized controlled trials, non-randomized studies

## Abstract

Reports suggest possible risks of adverse cardiovascular reactions, including heart failure, associated with non-ergot dopamine agonist (DA) use in Parkinson’s disease (PD). The objectives of our review were to evaluate the risk of heart failure and other adverse cardiovascular reactions in PD patients who received a non-ergot DA compared with other anti-PD pharmacological interventions, placebo, or no intervention. Studies were identified via searches of six bibliographic databases. Randomized controlled trials (RCTs) and non-randomized studies (NRS) were eligible for study inclusion. Random-effect meta-analyses were performed to estimate adverse cardiovascular reaction risks. Quality of evidence was assessed using GRADE. In total, forty-four studies (thirty-six RCTs and eight NRS) satisfied our inclusion criteria. A single RCT found no significant difference in the risk of heart failure with ropinirole compared with bromocriptine (odds ratio (OR) 0.39, 95% confidence interval (CI) 0.07 to 2.04; low certainty). Conversely, three case–control studies reported a risk of heart failure with non-ergot DA treatment. The quality of evidence for the risk of heart failure was judged as low or very low. Findings suggest that non-ergot DA use may be associated with adverse cardiovascular outcomes, including heart failure. Studies are needed to better understand cardiovascular risks associated with PD treatment.

## 1. Background

Parkinson’s disease (PD) is the second most common neurodegenerative disorder worldwide. It is characterized by the progressive degeneration of dopaminergic neurons in the substantia nigra pars compacta [1,2]. Globally, the incidence rate of PD is five to twenty-four cases per 100,000 individuals per year, with men having higher incidence rates than women [3,4]. The prevalence of PD in the general population ranges from 57 to 371 per 100,000 individuals. The disease is most common among older adults and is diagnosed in 1–2% of individuals over the age of 60 [5].

Early motor symptoms of PD generally include tremors, rigidity, bradykinesia, and difficulties walking and maintaining balance [6]. Individuals may also develop depression, sleep disorders, autonomic dysfunction (such as constipation, urinary dysfunction, and orthostatic hypotension), and cognitive impairment [7,8].

Despite there being no cure for PD, pharmacological and non-pharmacological interventions have proven effective at managing disease symptoms, thus contributing to improvements in quality of life. Routinely used medications in the treatment of PD include levodopa, dopamine receptor agonists (DAs), enzyme inhibitors (monoamine oxidase B and catechol-o-methyltransferase inhibitors), anticholinergics, and amantadine [7]. Levodopa is considered the gold standard for PD treatment; although potent, it is associated with many side effects, including nausea, dopa-induced chorea or dystonia, and hypotension [9,10]. Other anti-PD drugs are, therefore, often incorporated into the management of milder PD symptoms, including DAs [11].

Introduced in the 1970s, DAs are a class of medication indicated for the treatment of neurologic diseases, including PD and restless legs syndrome, as well as non-neurologic conditions such as hyperprolactinemia [12,13,14]. Dopamine agonists are categorized into two groups: (1) ergot derivatives (bromocriptine, cabergoline, pergolide, and lisuride) and (2) non-ergot derivatives (apomorphine, piribedil, pramipexole, ropinirole, and rotigotine) [10]. The medications act by stimulating dopamine receptors, though some DAs also stimulate non-dopaminergic receptors, such as α-adrenergic and serotonergic receptors [10]. Ergot-derived DAs stimulate both D1-like and D2-like receptors, while non-ergot DAs selectively stimulate D2-like receptors [10,15]. Evidence from in vitro and in vivo models suggests that non-ergot DAs may be neuroprotective; however, this remains to be demonstrated in clinical practice [16,17].

Adverse effects associated with DA use may be attributed to dopaminergic stimulation or idiosyncratic reactions [18] and include impulse control disorders, addiction/substance abuse, cognitive slowing/confusion, sleep attacks, cardiac valve fibrosis, pleuropulmonary fibrosis, retroperitoneal fibrosis, psychosis, nausea, vomiting, dizziness, postural hypotension, and peripheral edema [12,16,19].

Dopamine agonists, particularly ergot DAs, have received attention in recent decades for their association with adverse cardiovascular events other than orthostatic hypotension, such as valvulopathy and heart failure [20,21,22,23,24,25]. Increasing evidence of the risk of serious damage to heart valves resulted in the withdrawal of pergolide from markets in some countries, including Canada and the USA [26,27]. Although ergot DAs continue to be used globally, they are not recommended as a first-line pharmacological intervention for PD due to the risk of fibrotic reactions [12].

In recent years, safety concerns have shifted to risks of adverse cardiovascular reactions associated with non-ergot DAs, with two large non-randomized studies reporting an increased risk of heart failure following the use of pramipexole [21,24]. The USA Food and Drug Administration subsequently released a safety announcement in September 2012 advising the public of the potential risk of heart failure associated with pramipexole use [28].

The objective of our review was to evaluate the risk of heart failure and other adverse cardiovascular reactions in PD patients treated with non-ergot DAs, compared to other anti-PD pharmacological interventions, placebo, or no intervention.

## 2. Materials and Methods

Detailed methodology of our review may be found within the study protocol, which was registered in the National Institute for Health Research International Prospective Register of Systematic Reviews (PROSPERO) prior to the start of the study (ID: CRD42015029753). Minor deviations to our published review protocol are described in Appendix A. 

Studies eligible for inclusion in the review included randomized controlled trials (RCTs) and non-randomized studies (NRS), including cohort and case–control studies. Cross-over trials, case series, controlled before-and-after, quasi-randomized, and interrupted time-series studies were excluded. Eligible study participants had to have idiopathic PD, with no restrictions based on age, gender, disease duration, disease severity, or history of pre-existing cardiovascular disease. Interventions eligible for inclusion included any dose of a non-ergot DA (apomorphine, piribedil, pramipexole, ropinirole, or rotigotine). Eligible comparison groups had to have received other anti-PD pharmacological intervention(s), placebo, or no intervention. 

The primary outcome examined in this review was heart failure, as defined by individual studies. We also examined the following 11 secondary outcomes: blood pressure disorders (hypertension and hypotension), valvulopathy, pleural effusion, peripheral edema, myocardial infarction, arrhythmias, cardiovascular death, use of cardiac drugs (including angiotensin-converting enzyme inhibitors or adrenergic beta-antagonists), use of artificial pacemakers or implantable defibrillators, syncope, and stroke.

### 2.1. Identification of Eligible Studies

We searched the following databases to identify studies potentially eligible for inclusion in our review: the Cochrane Central Register of Controlled Trials (CENTRAL), MEDLINE, Embase, the Cumulative Index to Nursing and Allied Health (CINAHL), PsycINFO, and PubMed. Databases were searched up to 14 November 2019; the search was subsequently updated on 6 May 2022. The detailed search strategies for the bibliographic databases are provided in Appendix A. Additional sources, including reference lists of relevant articles and textbooks, conference proceedings, and trial registries, were searched to identify additional published and non-published relevant studies.

Two review authors (JAGC and YF) independently screened records identified using a two-stage process. First, the titles and abstracts identified through the database searches were screened for eligibility, followed by full-text review of retained studies. Disagreements were resolved through discussion or, if necessary, through input from a third review author. Study screening, as well as all data extraction and management, were completed using DistillerSR (Evidence Partners, Ottawa, ON, Canada). DistillerSR (https://www.distillersr.com/, accessed on 31 August 2022) is a comprehensive literature review software that offers duplicate reference detection, multiple reviewer screening, conflict resolution, documentation of exclusion reasons, risk of bias and quality assessments, and data extraction via custom forms.

### 2.2. Data Extraction

Two review authors (MH and PG) independently extracted data from included studies. Data were extracted on the following items: study participants, methods, pharmacological exposures and control interventions, outcome measures, and study results. For dichotomous outcomes, such as the occurrence or non-occurrence of heart failure, we reported measures of treatment effect as risk estimates (such as relative risks (RR), odds ratios (OR), hazard ratios (HR), and incidence rate ratios (IRR)) with 95% confidence intervals (CI).

In instances where key information was unclear or not reported in the study, we contacted the study authors. Extracted data were entered into Review Manager 5.3 (version 5.3.5, The Nordic Cochrane Centre, The Cochrane Collaboration, Copenhagen, Denmark).

### 2.3. Risk of Bias

Two review authors (MH and PG) independently assessed the risk of bias among included RCTs using the Cochrane risk of bias tool for randomized trials [29]. For cohort and case–control studies, study quality was assessed using the Newcastle–Ottawa quality assessment scale for observational studies [30]. The Newcastle–Ottawa quality assessment scale is provided in Appendix A.

### 2.4. Data Synthesis

We conducted random-effects meta-analyses to combine comparable data, including risk estimates (such as RRs, ORs, HRs, and IRRs) from RCTs and NRS separately to estimate the risk of primary and secondary outcomes. Heterogeneity was assessed using the I^2^ statistic, and reporting biases were evaluated by examining the symmetry of funnel plots.

We completed subgroup analyses, where possible, to evaluate the risk of heart failure and other adverse cardiovascular reactions with individual non-ergot DAs to determine whether our findings varied according to certain patient and intervention characteristics, such as age, sex, dose of non-ergot DA, PD disease duration, and pre-existing cardiovascular disease.

To assess the robustness of our findings, we conducted sensitivity analyses by (1) excluding RCTs rated as having a high risk of bias (in any of the assessed domains on the Cochrane risk of bias tool) from the meta-analyses and (2) excluding NRS rated as having a high risk of bias (score < 7 stars (of 9) on the Newcastle–Ottawa quality assessment scale) from the meta-analyses.

### 2.5. Assessment of the Certainty of the Evidence

We assessed the quality of the evidence across the studies identified for each outcome within each comparison using the GRADE approach [31,32]. The GRADE approach provides an overall quality rating across a body of evidence by assessing five factors that may lead to rating down: risk of bias, inconsistency, indirectness, imprecision, and publication bias. For NRS, three factors may lead to increased certainty within the outcome: large magnitude of effect, dose–response gradient, or plausible residual confounding. The GRADE assessments are displayed in the Summary of Findings table for each research question.

## 3. Results

### 3.1. Description of Studies

A total of 44 studies reporting on the risk of heart failure and other adverse cardiovascular reactions among patients with Parkinson’s disease (PD) following treatment with a non-ergot dopamine agonist (DA) (compared to other anti-PD pharmacological interventions, placebo, or no intervention) were included in our review. Of these studies, 36 were randomized controlled trials (RCTs), and eight were non-randomized studies (NRS) (four case–control studies and four cohort studies). A single RCT [33] and three case–control studies [21,24,34] reported data related to heart failure, our primary outcome, as an outcome of non-ergot DA treatment. All other studies reported associations between non-ergot DA exposure and other cardiovascular outcomes.

### 3.2. Results of the Search

Refer to Figure 1 for the study flow diagram. We identified a total of 1518 records in our 6 May 2022 search of electronic databases and four records by hand-searching and citation-searching relevant resources. After removing duplicate records, we screened the titles and abstracts of 1104 records for eligibility. Of these, the full text of 171 articles were retrieved and assessed for eligibility, with 44 distinct studies meeting our inclusion criteria.

### 3.3. Included Studies

Details of studies included in our review are provided in Appendix A. Of the 44 studies narratively described in our review, 32 studies were included in our quantitative synthesis (28 RCTs and four case–control studies). Of the remaining studies, eight RCTs were the sole report of a cardiovascular outcome or comparison of interest and, therefore, could not be pooled with results of other studies, while four cohort studies only reported counts of cardiovascular events.

With the exception of the study by Im et al. [35], all included RCTs were multicentre studies that recruited patients from a maximum of 98 centres located in Canada, the USA, Europe, South Africa, Latin America, China, and Japan. The majority of the RCTs were published after the year 2000, with only seven being published prior to that time [33,36,37,38,39,40,41]. Included RCTs varied in duration from 11 weeks to 4 years. The number of patients randomized in RCTs ranged from 30 to 535, with a mean of 284 patients (standard deviation (SD) = 149). Participants included both males and females, with the treated group’s mean age being between 57.5 and 79.2 years. The majority of recruited participants had a diagnosis of idiopathic PD with a mean disease duration ranging from 0.97 to 12.2 years. Eligibility criteria differed across RCTs. A multitude of inclusion criteria were applied, including but not limited to restrictions based on age, the severity of PD diagnosis (based on Hoehn and Yahr Staging Scale (H&Y) and/or Unified Parkinson’s Disease Rating Scale (UPDRS) scores), having uncontrolled motor symptoms, the minimum number of “off” hours per day, and/or time with PD. Patients were commonly excluded for a number of reasons, including but not limited to having dementia, psychiatric disorders, cardiovascular conditions (e.g., abnormal electrocardiogram, hypotension), or being treated with select medications. Included RCTs did not report any data for the following cardiovascular outcomes: (1) pleural effusion, (2) cardiovascular death, (3) use of cardiac drugs, and (4) use of artificial pacemakers or implantable defibrillators.

The eight included NRS were published between 2000 and 2016. The mean age of participants reported in these studies was between 59.9 years and 79.9 years. Among studies reporting mean duration with PD (four of eight), individuals had PD for approximately 3.2 to 11 years. The inclusion of subjects was most often anchored on their incident diagnosis of PD and/or if they were receiving PD medications. Three case–control studies [21,24,34] were analyses of large databases and included populations ranging from 14,122 to 26,814 patients over a period of 11 to 13 years. The smallest case–control study [42] included 210 Japanese patients and was completed at a single centre. Findings from all case–control studies were included in our quantitative synthesis. Four NRS included in our review were cohort studies with follow-up periods ranging from 6 to 21 months. All were single-centre studies [20,25,42,43] except for one study conducted in two hospitals [44]. The mean sample size was 219 patients (SD = 178), with a range of 102–527 patients. Findings reported by cohort studies were limited to count data of adverse cardiovascular events, including the occurrence of hypotension, heart failure, syncope, and/or valvulopathy following exposure to one or more non-ergot DAs (Appendix A). Included NRS did not report any data for the following cardiovascular outcomes: (1) hypertension, (2) pleural effusion, (3) peripheral edema, (4) arrhythmia, (5) cardiovascular death, (6) use of cardiac drugs, and (7) use of artificial pacemakers or implantable defibrillators.

Randomized controlled trials and NRS included in this review report on the occurrence of our primary or secondary cardiovascular outcomes for comparisons of interest, which included any dose of a non-ergot DA compared to other anti-PD pharmacological interventions(s), placebo, or no intervention. Among the included studies, one or more cardiovascular outcomes were reported for 24 distinct comparisons. Not all outcomes were reported for each comparison. The many eligible comparisons and outcomes resulted in a total of 74 unique comparison–outcome combinations being included in our review, all of which were eligible for inclusion in our quantitative synthesis. Of these, 57 comparison–outcome combinations were described by only one study. Individual comparison–outcome reports are detailed in Appendix A; however, these are not included in our meta-analyses of effects. Overall, we completed meta-analyses for 17 distinct comparison–outcome combinations across 10 comparisons and five outcomes. Treatment with pramipexole compared to placebo (or no treatment in the case of NRS) was the most commonly reported comparison of interest. Orthostatic hypotension was the most frequently reported outcome examined in our quantitative synthesis and was categorized as either (1) orthostatic hypotension (symptomatic and asymptomatic), (2) symptomatic hypotension, (3) asymptomatic hypotension, or (4) hypotension. The remaining outcomes included in our quantitative synthesis are (1) heart failure, (2) peripheral edema, (3) valvulopathy, and (4) hypertension.

Included RCTs report on 20 (83.3%) of the 24 distinct comparisons examined in our review and 51 (68.9%) of the 74 aforementioned comparison–outcome combinations (39 (76.5%) of which were described by a single study). Randomized controlled trials were included in meta-analyses for 12 of 17 (70.6%) distinct comparison–outcome combinations across eight comparisons and three outcomes. Cardiovascular effects following treatment with a form of pramipexole compared to placebo or other interventions were most commonly documented. Three cardiovascular outcomes were included in our meta-analyses of RCT data: (1) orthostatic hypotension, (2) peripheral edema, and (3) hypertension.

Included NRS report on approximately one-third of the comparisons (eight of 24, 33.3%) and comparison–outcome combinations (23 of 74, 31.1%; 18 (78.3%) of which were described by a single study) examined within our review. Non-randomized studies were included in meta-analyses for five of 17 (29.4%) distinct comparison–outcome combinations across four comparisons and two outcomes. Cardiovascular effects following treatment with pramipexole or ropinirole compared to no or current/past levodopa treatment were most frequently investigated. Two cardiovascular outcomes were included in our meta-analyses of NRS data: (1) heart failure and (2) valvulopathy.

In consideration of the large number of comparisons and few cardiovascular outcomes examined by our review, our qualitative synthesis is reported according to our study outcomes.

### 3.4. Excluded Studies

Details of the 104 studies that were excluded following full-text review, including specific reasons for exclusion, are provided in the list of excluded studies (Appendix A). Overall, studies were excluded for not satisfying the following inclusion criteria: population (6), intervention (6), comparison (28), outcome (12), study design (48), or duplicate citation (4).

### 3.5. Studies Awaiting Classification

Some potentially relevant studies identified through our search had insufficient information to apply our inclusion/exclusion criteria and remain awaiting classification pending the availability of additional information. The 23 studies awaiting classification are summarized in Appendix A. There were 15 studies potentially eligible for inclusion in our review identified via the updated search of all databases on 6 May 2022. These studies are not included in our qualitative or quantitative syntheses since none of the studies awaiting classification report on the risk of heart failure, our primary outcome, associated with non-ergot DA use for PD. Furthermore, many studies awaiting classification are solely reported as abstracts or are published in languages other than English. As of May 2024, review authors are unaware of any new evidence that would meaningfully alter the findings reported within this review.

### 3.6. Risk of Bias in Included Studies

The risk of bias in included studies is summarized in Figure 2 and Figure 3.

The following summarizes bias within included studies reporting on our primary outcome, heart failure. There were four studies that reported on heart failure and were included in our quantitative synthesis; one RCT [33] and three case–control studies [21,24,34]. The risk of bias in Korczyn et al. [33] was assessed for each domain, whereas the Newcastle–Ottawa quality assessment scale [30] was used to determine the quality of the aforementioned case–control studies.

We assessed the findings reported in the RCT by Korczyn et al. [33] to be subject to serious risk of bias since investigators reported limited or no information about sequence generation; allocation concealment; and blinding of participants, personnel, and outcome assessors. Refer to Appendix A for complete risk of bias assessment details.

Among the case–control studies reporting heart failure included in our meta-analyses [21,24,34], all were deemed to be of high quality (7+ of 9 possible stars on the Newcastle–Ottawa quality assessment scale), with selection, comparability, and exposure ratings being similar across all studies (Table 1). The study by Mokhles et al. [24] was rated as the highest quality (8 stars).

The risk of bias reported for secondary outcomes was assessed, as appropriate, and is summarized in the Characteristics of included studies table.

### 3.7. Effects of Interventions

#### 3.7.1. Primary Outcome

##### Heart Failure

We identified a single RCT that reported heart failure as an outcome or adverse event to the treatment of PD with non-ergot DAs [33]. Specifically, this 3-year trial of patients with early-stage PD found no significant difference in the risk of heart failure among patients treated with ropinirole compared to those treated with bromocriptine (odds ratio (OR) 0.39; 95% confidence interval (CI) 0.07–2.04; low certainty).

There were three case–control studies reporting heart failure as an outcome of PD treatment [21,24,34]. Collectively, four separate treatment comparisons were included in our random-effect meta-analyses: (1) the pooled effect estimate from two studies showed a moderate risk of heart failure (adjusted odds ratio (AOR) 1.46, 95% CI 1.03 to 2.08; low certainty) among patients treated with pramipexole compared to non-users of pramipexole [21,34] (Figure 4); (2) the pooled effect estimate from two studies showed a moderate risk of heart failure (AOR 1.54, 95% CI 1.21 to 1.98; low certainty) among patients treated with pramipexole compared to those treated with levodopa users (including current/past use) [24,34] (Figure 5); (3) the pooled effect estimate from two studies showed no increased risk of heart failure among patients treated with ropinirole compared to non-users ropinirole (AOR 1.04, 95% CI 0.87 to 1.24; very low certainty) [21,34] (Figure 6); and (4) the pooled effect estimate from two studies showed no increased risk of heart failure among patients treated with ropinirole compared to those treated with levodopa (including current/past use) (AOR 1.02, 95% CI 0.76 to 1.37; very low certainty) [24,34] (Figure 7). All three studies utilized administrative claims or electronic health record data to examine the risk of health failure among large cohorts of patients with idiopathic PD who received non-ergot DAs for the management of motor complications. Although subgroup analyses were not comparable across studies, the study by Mokhles et al. [24] reported a noteworthy time-specific risk of heart failure for new users of pramipexole. Specifically, investigators found an increased risk of heart failure in the immediate 3 months after initial pramipexole treatment (AOR 3.06, 95% CI 1.74 to 5.39; compared to non-use), which diminished with continued use beyond 3 months.

Finally, a large UK study (n = 26,814) of incident heart failure among individuals 40–89 years of age with PD reported no increased risk of heart failure for three separate non-ergot DA comparisons of interest: (1) pramipexole users compared to users of all other DAs (relative risk (RR) 1.28, 95% CI 0.82 to 2.00; very low certainty), (2) pramipexole users compared to ergot DA users (RR 1.07, 95% CI 0.66 to 1.74; very low certainty), and (3) pramipexole users compared to users of all other non-ergot DAs (RR 1.53, 95% CI 0.92 to 2.57; very low certainty) [21].

#### 3.7.2. Secondary Outcomes

Included studies reported findings for eight of 12 secondary cardiovascular outcomes of interest. No studies were found that assessed pleural effusion, cardiovascular death, use of cardiovascular drugs, or the use of artificial pacemakers/defibrillators resulting from the use of non-ergot DAs for PD. Findings from all of our meta-analyses for secondary outcomes are presented in Figure 8, Figure 9, Figure 10, Figure 11, Figure 12, Figure 13, Figure 14, Figure 15, Figure 16, Figure 17, Figure 18, Figure 19, Figure 20, Figure 21, Figure 22, Figure 23 and Figure 24. Findings reported by a single study for a particular comparison–outcome combination, and therefore not included in any meta-analysis, are described in Appendix A for RCTs and NRS, respectively. Results for cardiovascular outcomes most commonly reported (orthostatic hypotension, peripheral edema, and valvulopathy) within included studies are summarized below.

##### Orthostatic Hypotension

Twenty-eight RCTs reported orthostatic hypotension as an outcome of PD treatment. There were seven distinct treatment comparisons included in our meta-analyses: (1) pramipexole compared to placebo [36,41,45,59,63] (Figure 8); (2) ropinirole compared to placebo [40,54,58] (Figure 9); (3) rotigotine compared to placebo [52,54,57,59,69] (Figure 10); (4) piribedil compared to placebo [61,72] (Figure 11); (5) pramipexole immediate release (IR) compared to pramipexole extended release (ER) [53,63,68] (Figure 12); (6) ropinirole compared to bromocriptine [33,35] (Figure 13); and (7) apomorphine compared to placebo [48,50] (Figure 14). Compared to placebo, we found that patients treated with rotigotine had a significantly reduced risk of developing orthostatic hypotension (OR 0.39, 95% CI 0.22 to 0.68; moderate certainty). The remaining six comparisons did not show statistically significant associations with hypotension; the pooled OR ranged from 0.26 to 22.91.

Four RCTs examined the effects of pramipexole (compared to placebo) on diagnosed symptomatic [38,56,59,72] (Figure 15) or asymptomatic [38,55] (Figure 16) hypotension. Pooled estimates for these specific categories of hypotension did not show statistically significant effects.

##### Peripheral Edema

There were eight RCTs that reported peripheral edema among patients who received pramipexole [51,62,64] or rotigotine [46,52,54,57,69] for the treatment of PD. Compared to placebo, we found an increased risk of peripheral edema associated with pramipexole (OR 2.97, 95% CI 1.50 to 5.88; moderate certainty) (Figure 17) but not with rotigotine (OR 1.38, 95% CI 0.59 to 3.22; moderate certainty) (Figure 18). All RCTs were relatively large (242 to 535 patients randomized) multicentre (39 to 98 centres) trials with a treatment duration of up to 7 months.

##### Valvulopathy

Two case–control studies reported valvulopathy as an outcome of pramipexole treatment (compared to no pramipexole treatment) for PD [34,42]. Overall, there was no statistically significant increase in valvulopathy (AOR 1.13, 95% CI 0.88 to 1.45; very low certainty) among 1000 cases and 3044 controls (Figure 19). The study by Crispo and colleagues [34] contributed 96.2% of the weight to the pooled estimate and involved analysis of electronic health records collected between 2000 and 2012 in the USA. Conversely, the study by Yamamoto et al. [42] was a single-centre study involving 210 patients in Japan.

### 3.8. Subgroup Analyses

Subgroup analyses were only possible according to patient age and non-ergot DA dose. We did not perform planned subgroup analyses stratified by other factors since few included studies reported findings by these variables.

#### 3.8.1. Age

Associations between non-ergot DA exposure and cardiovascular outcomes by age were reported by one RCT that examined the effects of rotigotine on “off” time in advanced PD [57]. In an age-stratified analysis involving 514 patients, Nicholas and colleagues [57] examined the occurrence of specific adverse events, including orthostatic hypotension and peripheral edema, in rotigotine and placebo-treated (12-week maintenance) groups that were <75 years and ≥75 years of age, respectively. The overall pooled effect estimate for orthostatic hypotension was similar to that for each age group; no heterogeneity or significant differences were observed between subgroups in our meta-analysis (Figure 20). Overall, compared to placebo, findings for rotigotine demonstrate a protective effect against orthostatic hypotension among patients with advanced PD.

Our meta-analysis of findings reported by Nicholas et al. [57] for peripheral edema shows heterogeneity between age subgroups. Patients < 75 years of age treated with rotigotine had a non-significant increased risk of peripheral edema (OR 4.57, 95% CI 0.26 to 79.87) compared to patients treated with placebo. Patients ≥ 75 years of age treated with rotigotine were at a non-significant reduced risk of experiencing peripheral edema compared to placebo-treated counterparts (OR 0.26, 95% CI 0.05 to 1.39). Age subgroup differences in the risk of peripheral edema were not significant (Figure 21).

#### 3.8.2. Dose

Most studies titrated the optimal dose of non-ergot DAs within a pre-specified treatment range for each study participant; however, others reported the effects of fixed doses.

Two placebo-controlled RCTs [52,57] monitored participants for orthostatic hypotension as an adverse event to rotigotine treatment at various fixed doses (2 to 12 mg/24 h), with no discernible dose–response relationship detected in our meta-analysis of reported findings (Figure 10).

Peripheral edema was reported in three RCTs as an outcome of dose-specific treatment with either pramipexole [51] or rotigotine [46,52] compared to placebo. For examined non-ergot DA doses, our meta-analyses found no dose–response between treatment and the risk of peripheral edema (Figure 17 and Figure 18).

A single placebo-controlled RCT [57] assessed the risk of myocardial infarction and stroke among participants who received different fixed doses of rotigotine (0 to 8 mg/24 h). One myocardial infarction and stroke were reported in the placebo group; no events were reported among participants who received rotigotine in this study. As a result, no meaningful treatment effects could be estimated from these data (Figure 22 and Figure 23).

Lastly, a fixed-dose, dose–response RCT of prolonged-release (PR) ropinirole in patients with advanced PD noted hypertension as a common adverse event [70]. We found no statistically significant relationship between ropinirole dose and hypertension in our analysis (Figure 24).

### 3.9. Sensitivity Analysis

Sensitivity analyses were performed by (1) excluding RCTs that have a high risk of bias and (2) excluding NRS that have a high risk of bias. There were no meta-analyses in which RCT and NRS data were pooled; therefore, planned sensitivity analyses of combined data from these study designs were not possible.

Sensitivity analyses examining the robustness of associations between non-ergot DA exposure and our primary outcome, heart failure, were not possible since the corresponding NRS were all of high quality. For assessments of secondary outcomes, five distinct sensitivity analyses (Figure 25, Figure 26, Figure 27, Figure 28 and Figure 29) based on our reported meta-analyses (Figure 8, Figure 13, Figure 14, Figure 16 and Figure 19) were possible.

Four of the five sensitivity analyses were meta-analyses of RCT data, whereas a single sensitivity analysis included NRS data. All possible sensitivity analyses for RCT data reported orthostatic hypotension (either as ‘symptomatic and asymptomatic’ or ‘asymptomatic’) as an outcome following exposure to commonly prescribed non-ergot DAs, including pramipexole, ropinirole, or apomorphine. The sensitivity analysis involving NRS data assessed the effect of excluding a single study with a high risk of bias from our analysis examining the diagnosis of valvulopathy following treatment with pramipexole (relative to no use of pramipexole) for Parkinson’s disease.

Overall, the exclusion of studies with a high risk of bias from our main analyses did not meaningfully change the magnitude of observed effects nor the direction of estimated associations.

## 4. Discussion

### 4.1. Summary of Main Results

Prior primary studies have reported that dopamine agonists (DAs), which are commonly used in the treatment of Parkinson’s disease (PD), are associated with adverse cardiovascular outcomes [10,73]. Specifically, ergot DAs are known to cause heart valve disease [74], while the risk of heart failure with the use of non-ergot DAs has recently come into question [28]. We conducted a systematic review to summarize knowledge of cardiovascular risks associated with non-ergot DA use in PD. Our objective was to evaluate the risk of heart failure and other adverse cardiovascular reactions among PD patients treated with these drugs. Both randomized controlled trials (RCTs) and non-randomized studies (NRS) were eligible for inclusion in our review. There were no RCTs included in our review that examined heart failure, our primary outcome, in relation to non-ergot DA exposure. The main finding from our meta-analyses of NRS was an increased risk of heart failure (low quality) among patients treated with pramipexole (compared to either non-users of pramipexole or current/past users of levodopa). Notable secondary findings include the following: (1) patients treated with pramipexole (compared to placebo) are at increased risk of peripheral edema (moderate quality), and (2) patients treated with rotigotine (compared to placebo) have a reduced risk of developing orthostatic hypotension (moderate quality). Our findings do not support an association between ropinirole and any examined cardiovascular outcome. The few possible subgroup analyses did not identify any differences in observed treatment effects by age or medication dose. Sensitivity analyses, where possible, show that estimates of effect derived in the meta-analyses were minimally affected by bias in the included studies. Our findings are discussed in detail in the following paragraphs and are presented in the Summary of Findings tables (Appendix A).

### 4.2. Heart Failure

Heart failure was first reported to be associated with the use of pramipexole by the USA Food and Drug Administration (FDA) [28], which prompted further NRS on this topic [10]. The FDA subsequently conducted a review of phase two and three clinical trial data of pramipexole. They found that heart failure was more frequently diagnosed among Parkinsonian and restless legs syndrome patients treated with pramipexole compared to non-users [28]. Pharmacoepidemiological studies later demonstrated an increased risk of heart failure following the use of pramipexole for the treatment of PD [21,24,34,75,76]. The underlying biological mechanism of this adverse cardiovascular reaction remains to be elucidated.

Results of our review support an increased risk of heart failure with pramipexole but not with ropinirole. Dopamine agonists reduce peripheral resistance, increase salt and water excretion, reduce endothelial activity, and mitigate insulin resistance by activating dopaminergic and non-dopaminergic receptors [10]. However, only cabergoline and pramipexole have been connected with heart failure, the former by producing heart valve fibrosis after activation of 5-HT receptors. The potential mechanism for the increased risk of heart failure with pramipexole remains a mystery. The cardiovascular effects of dopamine agonists cannot fully account for the increased risk of heart failure, as ropinirole has the same effects and yet does not increase heart failure. Pramipexole, but not ropinirole, is excreted unmetabolized through the kidney. It can be hypothesized that it might alter the intra-renal renin–angiotensin system, which could produce renal inflammation and liquid retention, which might contribute to heart failure [77,78].

Our findings for heart failure are primarily based on data from NRS, which provides data under real-world conditions of use. Reporting of adverse cardiovascular reactions related to non-ergot DA use in RCTs is potentially limited by their size and duration, as well as the exclusion of populations at high risk for adverse outcomes.

### 4.3. Orthostatic Hypotension

It has been reported that orthostatic hypotension may affect approximately 20–65% of PD patients [79,80] and that it may result from the neurodegenerative process itself or extrinsic influences, including older age [81,82,83] and exposure to hypotensive drugs [81,82]. Moreover, hypotensive effects have been observed for some antiparkinsonian drugs [84], with studies showing that orthostatic hypotension occurs in patients treated with DAs [85,86,87] and is less frequent among levodopa users [85,88]. Monoamine oxidase B inhibitors have also been reported to induce orthostatic hypotension in PD patients [89]. These effects are likely attributed to direct or indirect activation of dopamine receptors, causing splanchnic vasodilation and renal salt and water loss [90,91,92]. Additionally, a sympatholytic effect, which may be related to alpha-adrenoreceptor partial agonism, can contribute to the observed effect [93].

In clinical trials of DAs, orthostatic hypotension was observed in 13.5% of the 8563 PD patients enrolled (15.4% in trials of ergot DAs compared to 11.3% in trials of non-ergot DAs) [18]. Similarly, no differences related to the prescribed type of DA were observed in spontaneous reports of orthostatic hypotension within the French Pharmacovigilance Database [94]. In our review, we compared the prevalence of orthostatic hypotension among groups of patients treated with non-ergot DAs. Most studies did not distinguish as to whether orthostatic hypotension was symptomatic or asymptomatic. Nevertheless, pooled effect estimates of orthostatic hypotension from our meta-analyses for pramipexole (compared to placebo) were similar across types of hypotension, with no increased or decreased risk being observed. Similar findings were found when comparing extended- and immediate-release formulations of pramipexole, as well as ropinirole or piribedil, with placebo. Unexpectedly, the pooled effect for orthostatic hypotension from RCTs was found to be lower among users of transdermal rotigotine compared to non-users, with heterogeneity being low and no dose–response relationship observed. These results are difficult to interpret in light of the current understanding of the pharmacodynamics of DAs [95]. It is tempting to hypothesize that pharmacokinetic differences, consisting of less abrupt changes in dopaminergic stimulation with transdermal rotigotine, may be related to the observed effect. No NRS examining the relationship between rotigotine exposure and orthostatic hypotension were included in our review. Additional research is therefore warranted in this area.

### 4.4. Peripheral Edema

Peripheral edema has been reported by PD patients using both ergot and non-ergot DAs [18], with 2.7% of more than 8,000 participants in RCTs of DAs experiencing this adverse outcome (3.1% among ergot DA users compared to 2.3% among non-ergot DA users) [18]. In an analysis of spontaneous reports to the French Pharmacovigilance Database, edemas were more frequently observed among ropinirole compared to levodopa users [94].

The mechanism of this side effect is unknown. This adverse drug reaction is sometimes reported as dose-dependent but may also be idiosyncratic. Risk factors for edema are female sex and cardiovascular comorbidities [96]. Discontinuation of the offending drug usually leads to edema reabsorption, and therefore, the use of a diuretic therapy is discouraged [96]. As expected, our meta-analysis comparing pramipexole use to no use found a statistically significant increased risk of peripheral edema. Conversely, there was no significant association between transdermal rotigotine exposure and peripheral edema. As with orthostatic hypotension, no NRS examining peripheral edema as an outcome of non-ergot DA exposure were included in our review. Future studies in this area are necessary.

### 4.5. Valvulopathy

Several reports suggest an association between the development of valvular dysfunction and exposure to ergot DAs, mainly pergolide and cabergoline [22,74,97]. The echocardiographic and histological features were very similar to those found in patients under fenfluramine or ergotamine or with carcinoid heart disease, thus suggesting a possible involvement of serotoninergic receptor activation [98]. The manufacturer voluntarily removed pergolide from the market in the USA (2007), while the European Medicines Agency issued a recommendation to limit the maximum dose of both pergolide and cabergoline to 3 mg and to monitor patients with regular echocardiograms. While our study did not focus on the effects of ergot DAs, our findings show no increased risk of valvulopathy with pramipexole.

### 4.6. Overall Completeness and Applicability of Evidence

There are a number of factors that limit the completeness and external validity of the findings reported by our systematic review. All studies included in our review directly examined the association between non-ergot DA exposure and the risk of one or more cardiovascular outcomes. We did not identify any study that reported on pleural effusion, cardiovascular death, use of cardiovascular drugs, or use of artificial pacemakers/defibrillators as an outcome of non-ergot DA use for PD. Therefore, we were unable to make any conclusions regarding the association between non-ergot DA use and these outcomes.

Since we relied on cardiovascular outcome definitions reported by study investigators, it is possible that definitions for select outcomes, such as valvulopathy (defined as valve-specific regurgitation, valve thickness, or valve calcification), examined within our review varied across studies. The same also applies to exposure ascertainment in NRS. Underreporting of adverse cardiovascular outcomes may have occurred irrespective of study type, especially in the context of RCTs where the occurrence of adverse events was sometimes only reported above a predefined threshold (such as 1% or 5%).

We were unable to directly compare cardiovascular safety findings from RCTs with those derived from NRS since meta-analyses for outcomes within each comparison included only RCT or NRS data. Within examined RCTs, risks of adverse cardiovascular outcomes may have been underestimated due to the inclusion of patients who are not representative of the general PD population. Although patients included in NRS may be more representative of real-world PD populations, reported findings within may be biased from exposure/outcome misclassification and residual or unmeasured confounding.

Collectively, the generalizability of our findings may be limited, especially among subpopulations with PD who were underrepresented within included studies (such as individuals with early onset disease). Nevertheless, our study identified moderate cardiovascular risks associated with the use of non-ergot DAs for PD, which warrant further investigation.

### 4.7. Quality of the Evidence

Refer to the risk of bias in the included studies tables and the ‘Risk of bias summary’ tables (Figure 2 and Figure 3). Across included RCTs, outcomes for examined comparisons were supported by very low-quality to high-quality evidence. Conversely, the outcomes of NRS included in our review were all supported by very low-quality evidence, with the exception of two comparisons (pramipexole compared to no treatment and pramipexole compared to levodopa) for heart failure that were supported by low-quality evidence. The quality of evidence for the risk of heart failure, our primary outcome, associated with non-ergot DA use for PD is summarized below.

We found a single RCT that examined heart failure diagnosis among patients with early-stage PD treated with either ropinirole or bromocriptine, with no significant risk identified. This was a large (37 centres), multinational study of 335 patients (ropinirole = 168; bromocriptine = 167) who were 30+ years of age. Overall, evidence of heart failure reported by authors was determined to be of low quality. We downgraded this study for imprecision and a serious risk of bias (Appendix A). This is an older study [33] for which we were unable to retrieve a trial protocol. As a result, the cumulative risk of bias was assessed as being serious since limited information on blinding methods, randomization, and allocation concealment was available from the published report of study findings. Moreover, for the same reason, it was unclear as to whether selective outcome reporting occurred. The 95% CI for the odds of heart failure suggested the potential for both harm and benefit with ropinirole treatment (compared to bromocriptine), therefore resulting in downgrading for imprecision.

There were three case–control studies included in our review that examined heart failure as an outcome of non-ergot DA treatment for PD. Due to the observational design of these studies, the quality of evidence for heart failure reported for each examined comparison by these studies was initially classified as low-quality according to the GRADE framework. Evidence for comparisons of pramipexole to no treatment and levodopa treatment, respectively, was not further upgraded or downgraded. For each comparison, pooled risk estimates suggested a moderate statistically significant risk of heart failure. On the other hand, evidence for comparisons of ropinirole to no treatment and levodopa treatment, respectively, was further downgraded to very low quality because the pooled estimates for heart failure demonstrated imprecise results, with estimates including potential harms and benefits.

### 4.8. Potential Biases in the Review Process

We completed our review in accordance with the guidelines outlined in the Cochrane Handbook for Systematic Reviews of Interventions.

With input and assistance from medical research librarians, we systematically searched six bibliographic databases for relevant records. Our searches encompassed all dates and languages. In instances when required safety data were not reported in a particular study or when full texts corresponding to identified abstracts could not be located, we searched for the information in clinical trial registries or contacted corresponding authors. Moreover, all screening, data extraction, and quality assessments were completed independently by two review authors. Additionally, pilot data extraction forms were validated using a 10% sample of included studies prior to their use. Disagreements were resolved by consensus and with the assistance of a third party, where necessary.

We identified a few limitations to our review process. Due to many interventions and outcomes being examined in this study, we included specific adverse cardiovascular outcomes and employed validated harm filters in our electronic searches of the literature. As with other harm reviews, the search results may have omitted some studies that reported harm as a secondary outcome. It is also possible that some relevant studies retrieved by our searches were unintentionally excluded as a result of the inadequate reporting of all outcomes and adverse events. Therefore, it is plausible that some of our reported associations may be subject to bias due to selective outcome reporting or missing data. The effects of this bias were mitigated by our supplementary hand searches of trial registries and conference proceeding databases. Moreover, JAGC and YF reviewed the full text of efficacy studies that were believed (based on the abstract) to have also examined adverse events.

A total of 23 studies were not included in our review: four studies published in foreign languages that we were unable to assess according to our inclusion/exclusion criteria; eight study abstracts that were deemed to meet our inclusion criteria based on a review of available published materials; and 11 full-texts eligible studies that were identified during our updated search of the literature (see Characteristics of studies awaiting classification). These studies may be assessed for eligibility and inclusion in a future update of this review, though they are not anticipated to considerably impact our primary or secondary findings due to the various non-ergot dopamine agonist exposures and distinct cardiovascular outcomes reported within these studies. It is possible that our risk assessment findings for select secondary outcomes (such as valvulopathy and edema) are biased as a result of not being able to include the results of all scientific investigations in our analyses.

### 4.9. Agreements and Disagreements with Other Studies or Reviews

There are few systematic reviews reporting on the cardiovascular safety of non-ergot DAs for PD. Consistent with our findings, a 2015 systematic review of NRS found pramipexole use for PD to be associated with an increased risk of heart failure [99].

Not surprisingly, orthostatic hypotension was a frequently explored outcome in these studies, as it is easily diagnosed in clinical trials. In the review by Baker et al. [100] on RCTs of ergot and non-ergot DAs as monotherapy or adjunct therapy (compared to placebo) for the early treatment of PD, no significant difference in the risk of orthostatic hypotension was reported (OR 1.43, 95% CI 0.84 to 2.44). Similarly, compared to placebo, pramipexole was not found to increase the risk of orthostatic hypotension in advanced PD (OR 1.44, 95% CI 0.96 to 2.17) [101]. This is consistent with findings reported for both ropinirole and pramipexole in early and advanced PD [102,103,104]. Our findings also coincide with meta-analyses reporting the effects of long-acting formulations of non-ergot DAs on blood pressure [105] and dizziness (a symptom of orthostatic hypotension) among users of IR and ER formulations of pramipexole [106].

A prior Cochrane review found that, relative to placebo, PD treatment with pramipexole was associated with an increased risk of peripheral edema, which is supported by the finding of our meta-analysis [104].

Meta-analyses comparing ergot DA users to non-ergot DA users for PD report an increased risk of valvular heart disease [99,107], while another systematic review failed to find any study reporting an increased risk of heart valve disease among users of non-ergot medications for PD [22].

## 5. Authors’ Conclusions

### 5.1. Implications for Practice

Despite few non-randomized studies (NRS) on the cardiovascular safety of non-ergot dopamine agonists (DAs) and very low quality of available evidence, our results suggest that non-ergot DAs may be associated with select cardiovascular adverse outcomes, including heart failure and peripheral edema. Overall, these adverse outcomes often affect a large number of Parkinson’s disease (PD) patients. Therefore, physicians should remain attentive to the occurrence of these events. High-risk groups remain to be identified, but women and patients with cardiovascular comorbidities may benefit from close monitoring for edema. Similar benefits may be realized by monitoring for heart failure in older patients with cardiovascular comorbidities.

We found a significantly reduced risk of orthostatic hypotension among users of transdermal rotigotine for PD (compared to placebo). This suggests that orthostatic hypotension may not be a major issue with non-ergot DAs in the treatment of PD.

### 5.2. Implications for Research

The cardiovascular safety of non-ergot DAs has been insufficiently studied thus far. This view is reinforced by the small number of outcomes reported in the limited number of studies to date.

Cardiovascular adverse events such as heart failure may be underreported in clinical trials of non-ergot DAs. The importance of the continuous assessment of adverse events in clinical trials is highlighted by the identification of heart failure with pramipexole by the USA Food and Drug Administration (FDA) [28]. Should clinical trial data be made more readily available to the scientific community in the future, independent risk assessments and testing of additional scientific hypotheses would be possible.

It is important to appreciate that some adverse cardiovascular reactions may not be identified in clinical trials due to their long latency periods and/or very low frequency. Since DAs are known to affect cardiovascular function, such adverse reactions are anticipated to occur at some point in time. In these cases, active pharmacovigilance using available administrative claims and electronic health record data is recommended.

Finally, our meta-analysis of the data from five randomized controlled trials (RCTs) suggests that transdermal rotigotine users may experience a reduced risk of orthostatic hypotension (compared to placebo). This novel finding is difficult to explain. Nevertheless, the quality of the evidence for this association was moderate, thus supporting a likely effect. Interestingly, similar results were observed in select trials that were not included in our review. For example, a post hoc analysis of RCTs showed a reduced risk of orthostatic hypotension when rotigotine was used as an add-on treatment to selegiline-treated patients (compared to placebo) [108]. This effect should be explored in further RCTs, as there are few effective treatments for orthostatic hypotension in PD.

## Figures and Tables

**Figure 1 brainsci-14-00776-f001:**
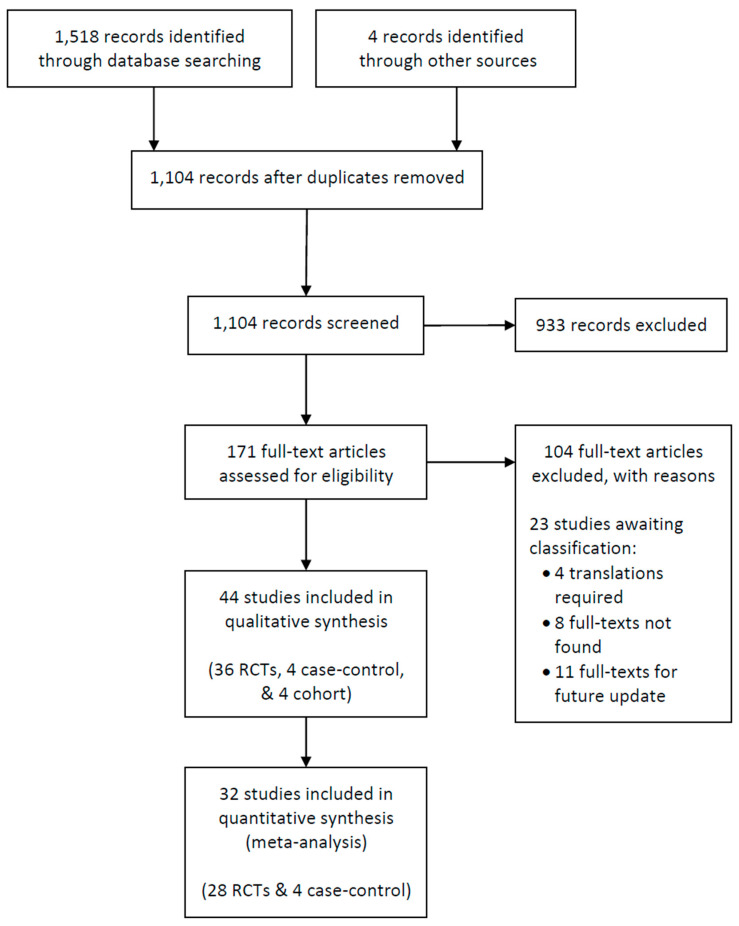
Study flow diagram.

**Figure 2 brainsci-14-00776-f002:**
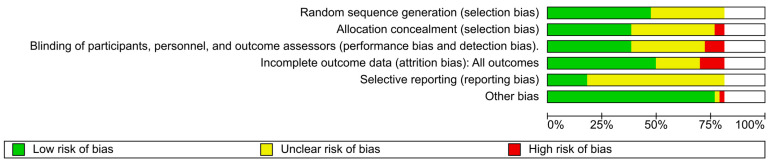
Risk of bias graph: review authors’ judgments about each risk of bias item presented as percentages across all included studies.

**Figure 3 brainsci-14-00776-f003:**
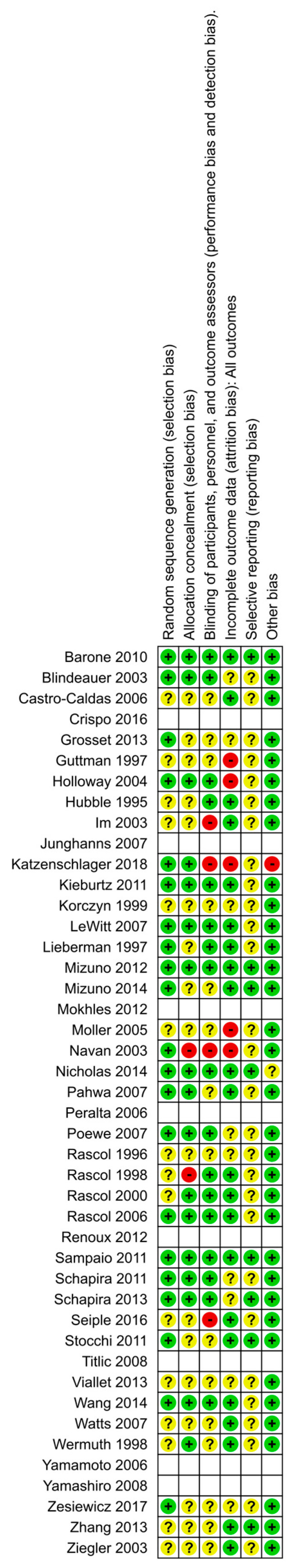
Risk of bias summary for randomized controlled trials: review authors’ judgments about each risk of bias item for each included study. Notes: Assessments of bias across examined domains were identical for each outcome reported within a study. Studies without ratings for all domains represent non-randomized studies. Ratings: (+) Low risk; (?) Some concerns; (-) High risk. Barone, 2010 [45]; Blindeauer, 2003 [46]; Castro-Caldas, 2006 [47]; Crispo, 2016 [34]; Grosset, 2013 [48]; Guttman 1997 [36]; Holloway, 2004 [49]; Hubble, 1995 [37]; Im, 2003 [35]; Junghanns, 2007 [20]; Katzenschlager, 2018 [50]; Kieburtz, 2011 [51]; Korczyn, 1999 [33]; LeWitt, 2007 [52]; Lieberman, 1997 [38]; Mizuno, 2012 [53]; Mizuno, 2014 [54]; Moller, 2005 [55]; Mokhles, 2012 [24]; Navan, 2003 [56]; Nicholas, 2014 [57]; Pahwa, 2007 [58]; Peralta, 2006 [25]; Poewe, 2007 [59]; Rascol, 1996 [40]; Rascol, 1998 [39]; Rascol, 2000 [60]; Rascol, 2006 [61]; Renoux, 2012 [21]; Sampaio, 2011 [62]; Schapira, 2011 [63]; Schapira, 2013 [64]; Seiple, 2016 [65]; Stocchi, 2011 [66]; Titlic, 2008 [44]; Viallet, 2013 [67]; Wang, 2014 [68]; Watts, 2007 [69]; Wermuth, 1998 [41]; Yamamoto, 2006 [42]; Yamashiro, 2008 [43]; Zesiewicz, 2017 [70]; Zhang, 2013 [71]; Ziegler, 2003 [72].

**Figure 4 brainsci-14-00776-f004:**
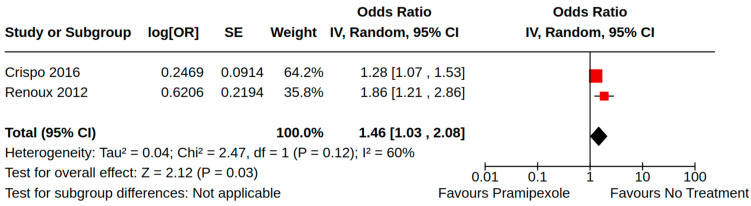
Forest plot of comparison: NRS pramipexole vs. no pramipexole. Outcome: Heart failure. Crispo, 2016 [34]; Renoux, 2012 [21].

**Figure 5 brainsci-14-00776-f005:**
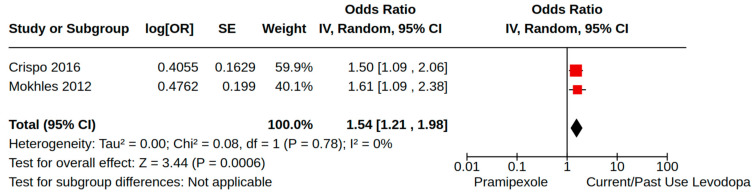
Forest plot of comparison: NRS pramipexole vs. current/past use levodopa. Outcome: Heart failure. Crispo, 2016 [34]; Mokhles, 2012 [24].

**Figure 6 brainsci-14-00776-f006:**
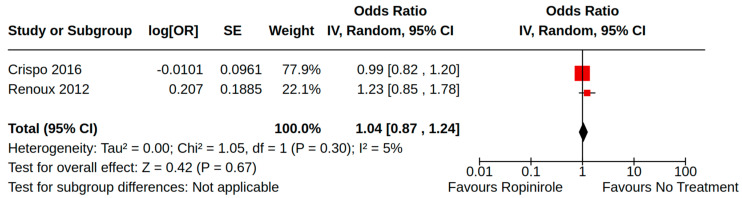
Forest plot of comparison: NRS ropinirole vs. no ropinirole. Outcome: Heart failure. Crispo, 2016 [34]; Renoux, 2012 [21].

**Figure 7 brainsci-14-00776-f007:**
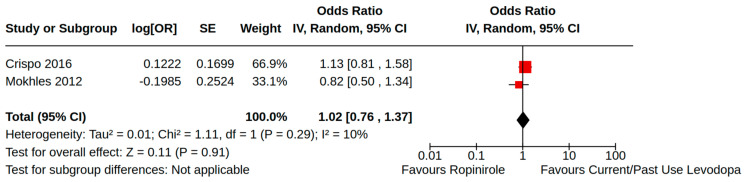
Forest plot of comparison: NRS ropinirole vs. current/past use levodopa. Outcome: Heart failure. Crispo, 2016 [34]; Mokhles, 2012 [24].

**Figure 8 brainsci-14-00776-f008:**
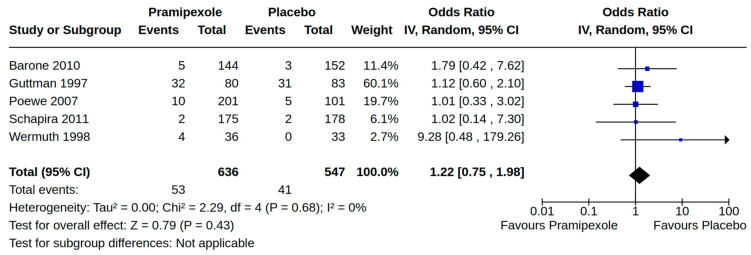
Forest plot of comparison: RCT pramipexole vs. placebo. Outcome 2: Orthostatic hypotension (symptomatic and asymptomatic). Barone, 2010 [45]; Guttman, 1997 [36]; Poewe, 2007 [59]; Schapira, 2011 [63]; Wermuth, 1998 [41].

**Figure 9 brainsci-14-00776-f009:**
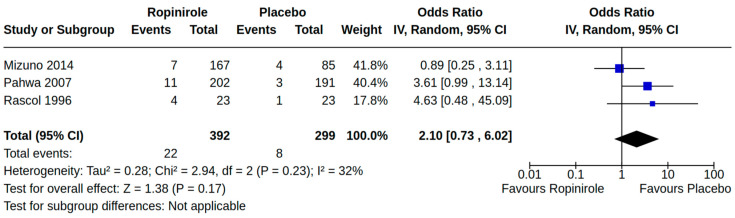
Forest plot of comparison: RCT ropinirole vs. placebo. Outcome 1: Orthostatic hypotension (symptomatic and asymptomatic). Mizuno, 2014 [54]; Pahwa, 2007 [58]; Rascol, 1996 [40].

**Figure 10 brainsci-14-00776-f010:**
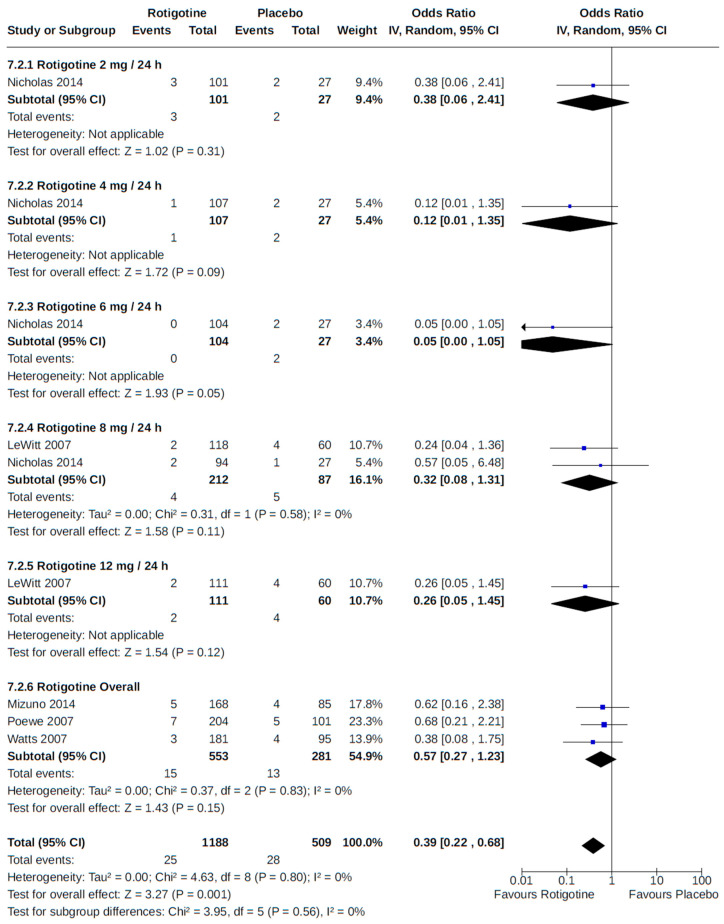
Forest plot of comparison: RCT rotigotine vs. placebo. Outcome 2: Orthostatic hypotension (symptomatic and asymptomatic). Nicholas, 2014 [57]; LeWitt, 2007 [52]; Mizuno, 2014 [54]; Poewe, 2007 [59]; Watts, 2007 [69].

**Figure 11 brainsci-14-00776-f011:**
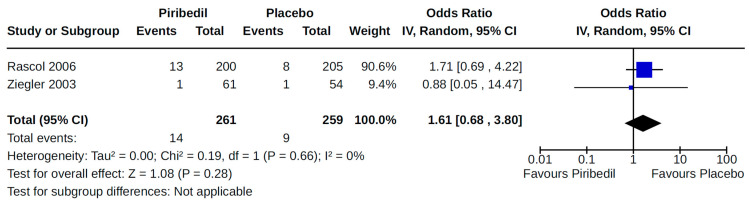
Forest plot of comparison: RCT piribedil vs. placebo. Outcome 1: Orthostatic Hypotension (symptomatic and asymptomatic). Rascol, 2006 [61]; Ziegler, 2003 [72].

**Figure 12 brainsci-14-00776-f012:**
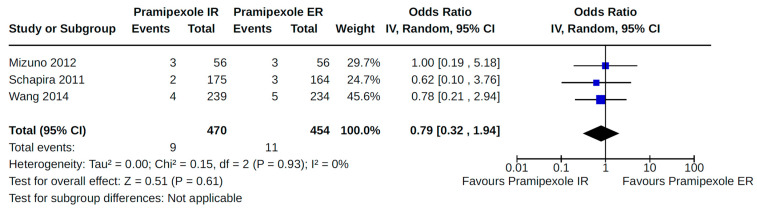
Forest plot of comparison: RCT pramipexole IR vs. pramipexole ER. Outcome 1: Orthostatic hypotension (symptomatic and asymptomatic). Mizuno, 2012 [53]; Schapira, 2011 [63]; Wang, 2014 [68].

**Figure 13 brainsci-14-00776-f013:**
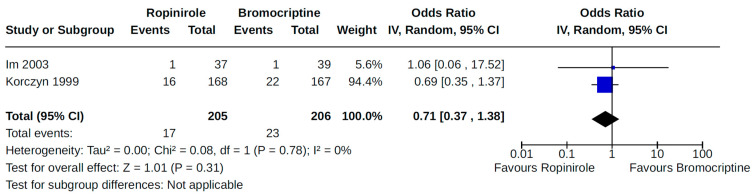
Forest plot of comparison: RCT ropinirole vs. bromocriptine. Outcome 1: Orthostatic hypotension (symptomatic and asymptomatic). Im, 2003 [35]; Korczyn, 1999 [33].

**Figure 14 brainsci-14-00776-f014:**
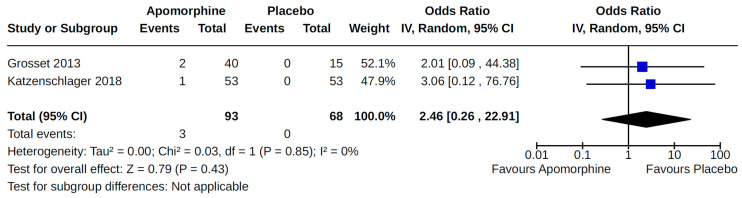
Forest plot of comparison: RCT apomorphine vs. placebo. Outcome 1: Orthostatic hypotension (symptomatic and asymptomatic). Grosset, 2013 [48]; Katzenschlager, 2018 [50].

**Figure 15 brainsci-14-00776-f015:**
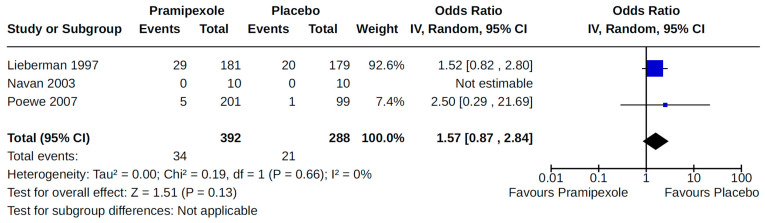
Forest plot of comparison: RCT pramipexole vs. placebo. Outcome 3: Symptomatic orthostatic hypotension. Lieberman, 1997 [38]; Navan, 2003 [56]; Poewe, 2007 [59].

**Figure 16 brainsci-14-00776-f016:**
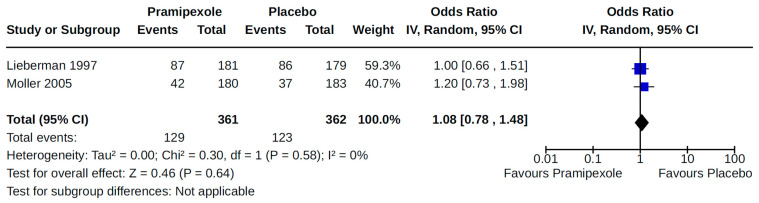
Forest plot of comparison: RCT pramipexole vs. placebo. Outcome 4: Asymptomatic orthostatic hypotension. Lieberman, 1997 [38]; Moller, 2005 [55].

**Figure 17 brainsci-14-00776-f017:**
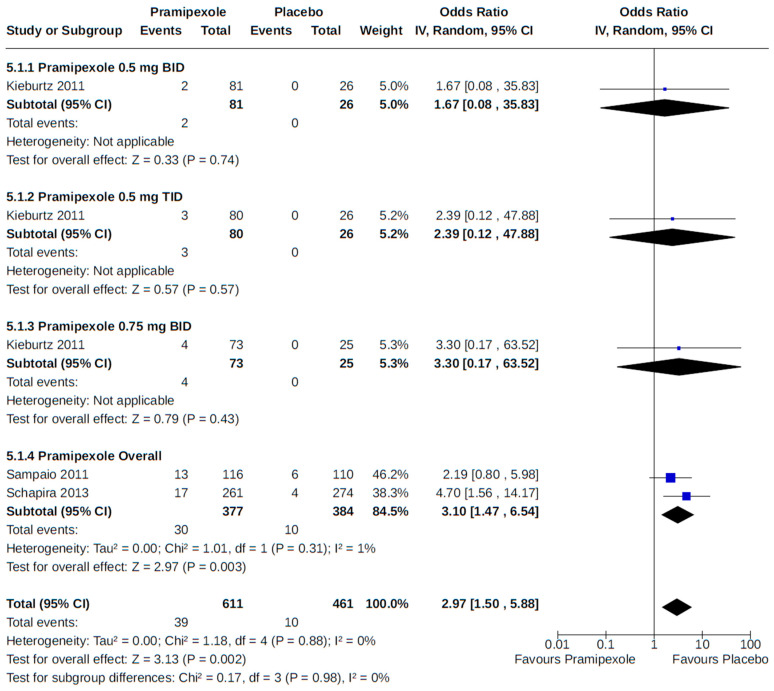
Forest plot of comparison: RCT pramipexole vs. placebo. Outcome 1: Peripheral edema. Kieburtz, 2011 [51]; Sampaio, 2011 [62]; Schapira, 2013 [64].

**Figure 18 brainsci-14-00776-f018:**
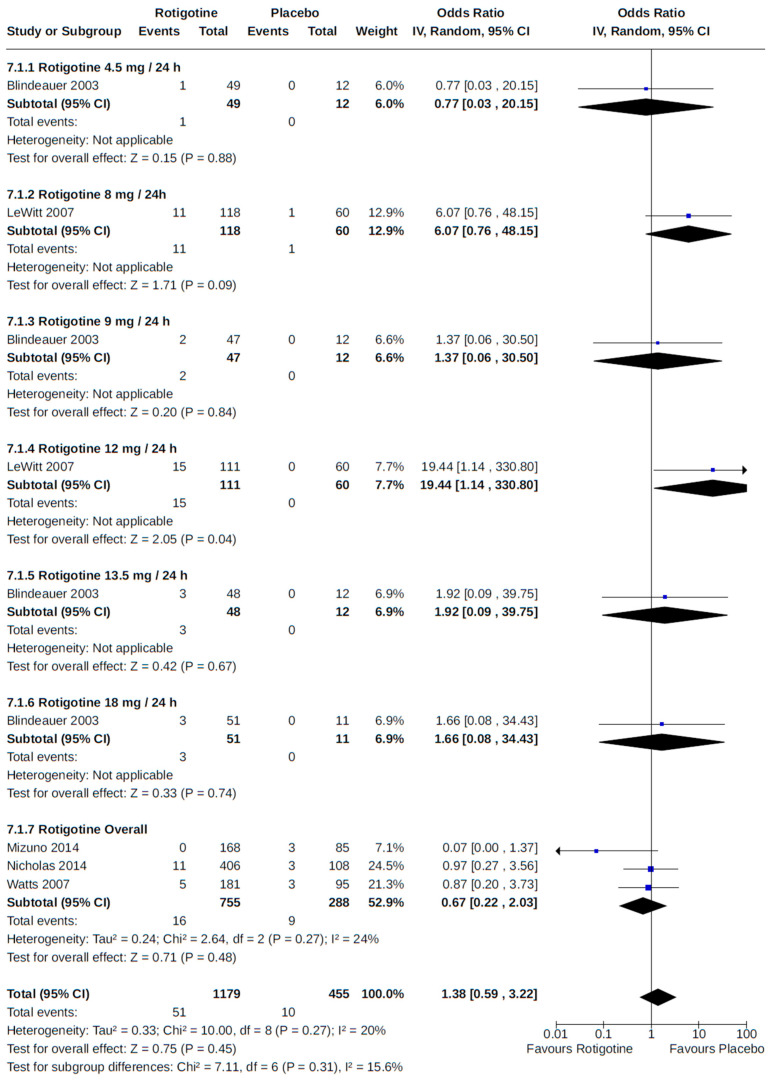
Forest plot of comparison: RCT rotigotine vs. placebo. Outcome 1: Peripheral edema. Blindeauer, 2003 [46]; LeWitt, 2007 [52]; Mizuno, 2014 [54]; Nicholas, 2014 [57]; Watts, 2007 [69].

**Figure 19 brainsci-14-00776-f019:**
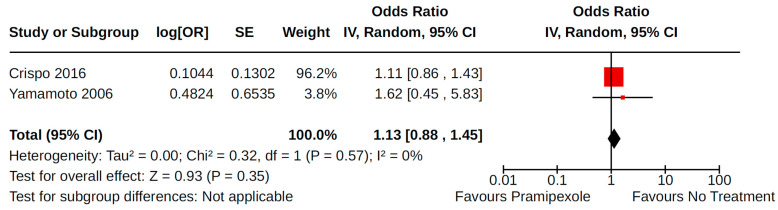
Forest plot of comparison: NRS pramipexole vs. no pramipexole. Outcome 2: Valvulopathy. Crispo, 2016 [34]; Yamamoto, 2006 [42].

**Figure 20 brainsci-14-00776-f020:**
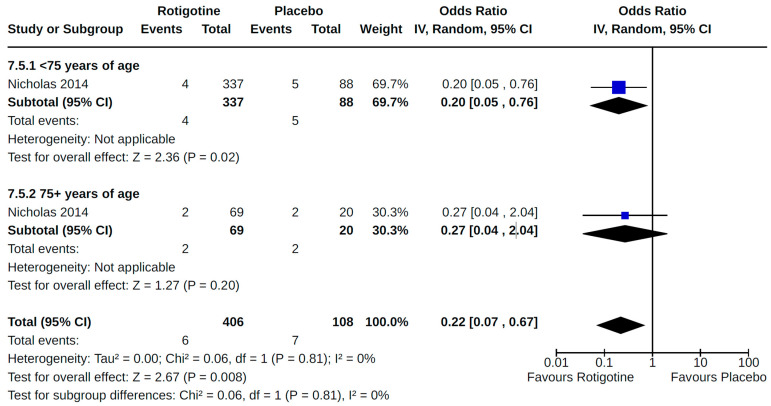
Forest plot of comparison: RCT rotigotine vs. placebo. Outcome 5: Orthostatic hypotension (symptomatic and asymptomatic)—by age. Nicholas, 2014 [57].

**Figure 21 brainsci-14-00776-f021:**
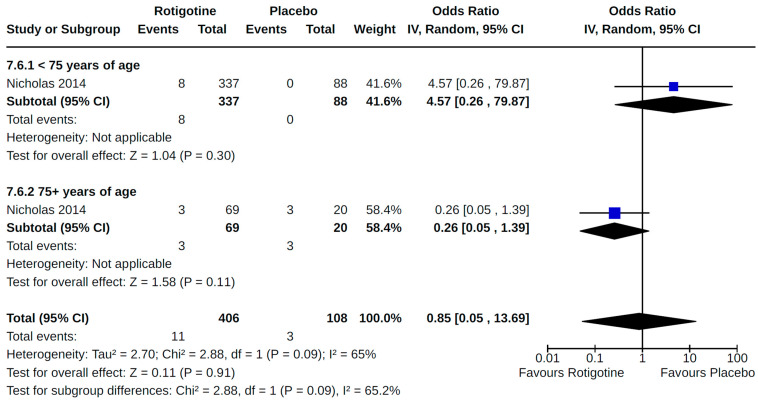
Forest plot of comparison: RCT rotigotine vs. placebo. Outcome 6: Peripheral edema—by age. Nicholas, 2014 [57].

**Figure 22 brainsci-14-00776-f022:**
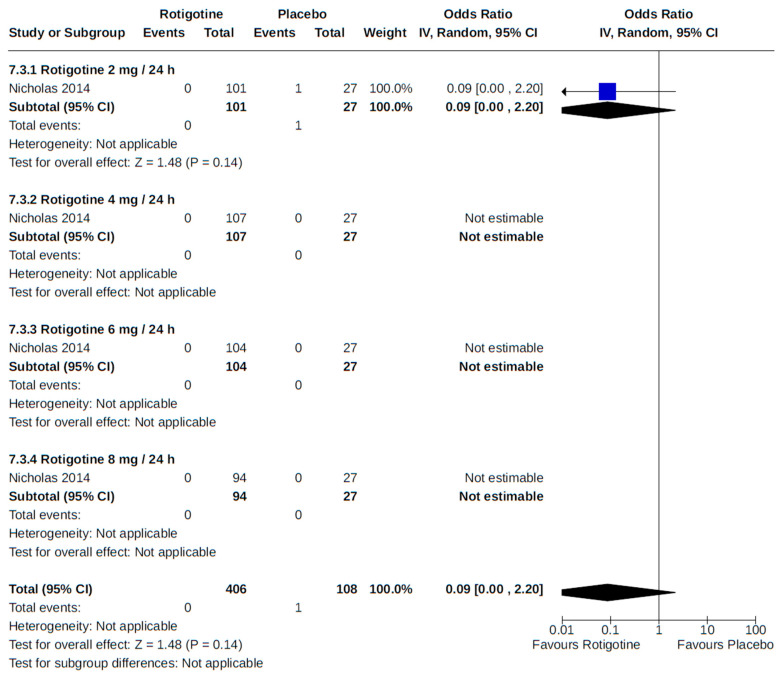
Forest plot of comparison: RCT rotigotine vs. placebo. Outcome 3: Myocardial infarction. Nicholas, 2014 [57].

**Figure 23 brainsci-14-00776-f023:**
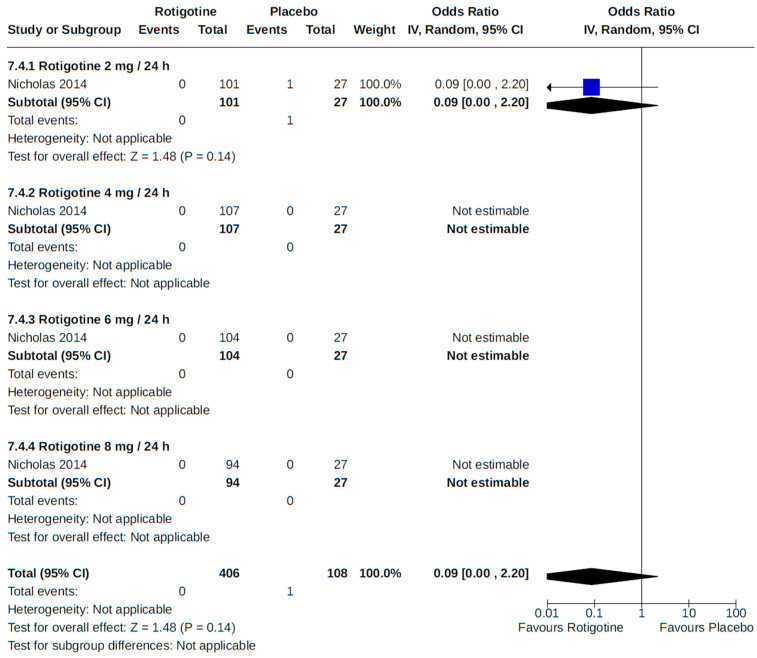
Forest plot of comparison: RCT rotigotine vs. placebo. Outcome 4: Stroke. Nicholas, 2014 [57].

**Figure 24 brainsci-14-00776-f024:**
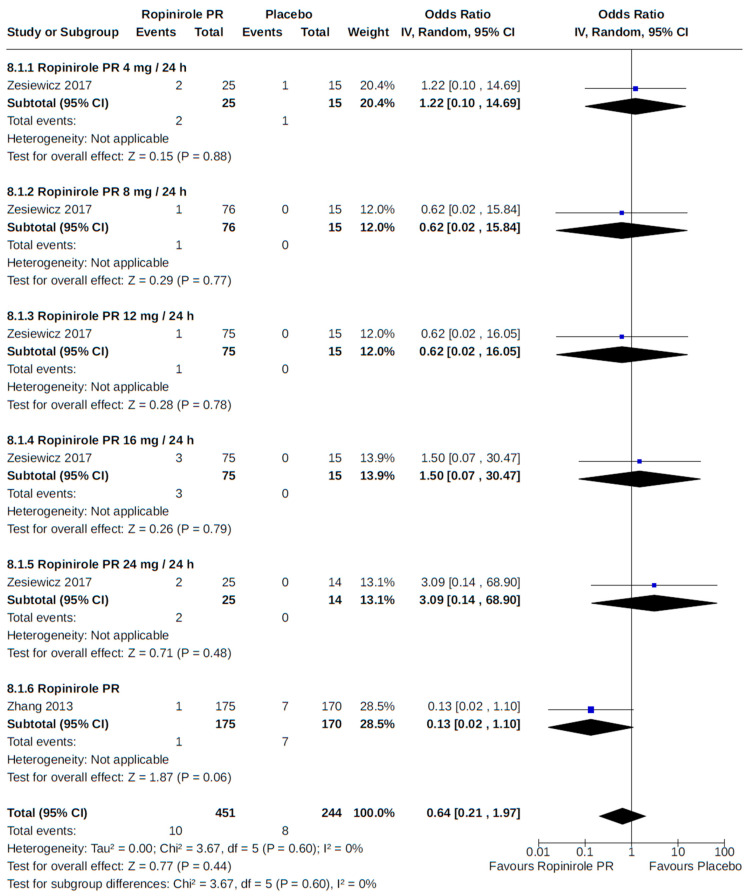
Forest plot of comparison: RCT ropinirole PR vs. placebo. Outcome 1: Hypertension. Zesiewicz, 2017 [70]; Zhang, 2013 [71].

**Figure 25 brainsci-14-00776-f025:**
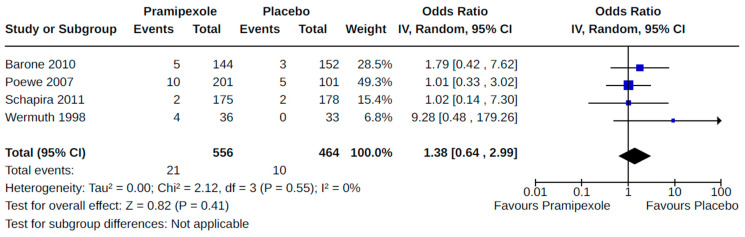
Sensitivity analyses. Outcome 1: Orthostatic hypotension (symptomatic and asymptomatic). Barone, 2010 [45]; Poewe, 2007 [59]; Schapira, 2011 [63]; Wermuth, 1998 [41].

**Figure 26 brainsci-14-00776-f026:**
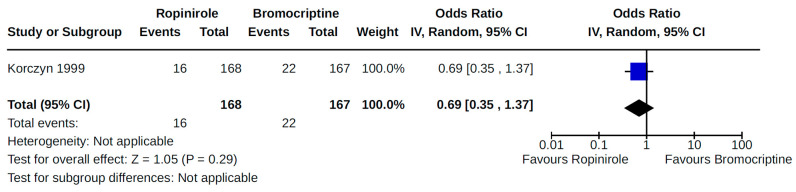
Sensitivity analyses. Outcome 3: Orthostatic hypotension (symptomatic and asymptomatic). Korczyn, 1999 [33].

**Figure 27 brainsci-14-00776-f027:**
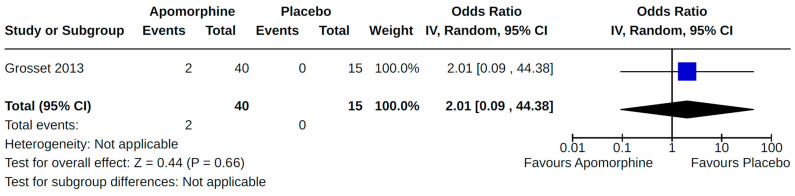
Sensitivity analyses. Outcome 4: Orthostatic hypotension (symptomatic and asymptomatic). Grosset, 2013 [48].

**Figure 28 brainsci-14-00776-f028:**
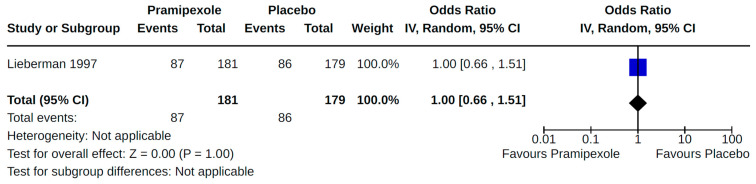
Sensitivity analyses. Outcome 2: Asymptomatic orthostatic hypotension. Lieberman, 1997 [38].

**Figure 29 brainsci-14-00776-f029:**
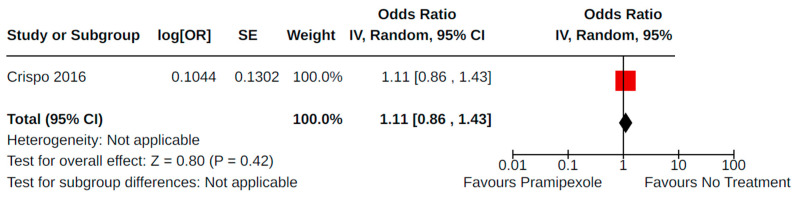
Sensitivity analyses. Outcome 5: Valvulopathy. Crispo, 2016 [34].

**Table 1 brainsci-14-00776-t001:** Newcastle–Ottawa Scores for the non-randomized studies. (Asterisks represent the number of stars awarded to a study within each category; refer to Appendix A for the detailed quality assessment tool).

Reference	Outcome	Selection	Comparability	Exposure
Crispo, 2016 [34]	Heart Failure	***	*	***
	Hypotension	***	*	***
	Valvulopathy	***	*	***
	Myocardial Infarction	***	*	***
	Stroke	***	*	***
Mokhles, 2012 [24]	Heart Failure	****	*	***
Renoux, 2012 [21]	Heart Failure	***	*	***
Yamamoto, 2006 [42]	Valvulopathy	***	**	

## Data Availability

The original contributions presented in this study are included in the article and Appendix A. Further inquiries can be directed to the corresponding author.

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
