# Peer review of "Non-Ergot Dopamine Agonists and the Risk of Heart Failure and Other Adverse Cardiovascular Reactions in Parkinson’s Disease"

_brainsci, 2024, doi:10.3390/brainsci14080776_

Round 1
Reviewer 1 Report
Comments and Suggestions for Authors
The submitted systemativ review is an important source of information concerning the risk of heart failure and other adverse cardiovascular reactions in PD patients who received a non-ergot DA, compared with other anti-PD pharmacological interventions, placebo, or no intervention.
The methods are appropriate and described in detail. The work was registered, as required. The data are interesting and indicate that non-ergot DA use may be associated with cardiovascular adverse outcomes, including heart failure.
Author Response
Comment 1: The submitted systematic review is an important source of information concerning the risk of heart failure and other adverse cardiovascular reactions in PD patients who received a non-ergot DA, compared with other anti-PD pharmacological interventions, placebo, or no intervention.
The methods are appropriate and described in detail. The work was registered, as required. The data are interesting and indicate that non-ergot DA use may be associated with cardiovascular adverse outcomes, including heart failure.
Response 1: We thank the Reviewer for their in-depth review of our systematic review, and for their feedback. We appreciate the Reviewer’s acknowledgement of our rigorous study methodology, as well as the novelty of our findings pertaining to the association between non-ergot dopamine agonist use and adverse cardiovascular reactions in Parkinson’s disease.
Reviewer 2 Report
Comments and Suggestions for Authors
The Manuscript (ID: brainsci-3087225) entitled “Non-ergot dopamine agonists and the risk of heart failure and other adverse cardiovascular reactions in Parkinson's disease” evaluated an interesting topic on the treatment of Parkinson’s disease. The aims of this review were to evaluate the risk of heart failure and other adverse cardiovascular reactions in PD patients who received a non-ergot dopamine agonist, compared with other anti-PD pharmacological treatments, placebo, or no intervention. Results indicated increased risk of heart failure in patients treated with pramipexole compared to other interventions.
In general, the Manuscript is well written and clear to understand, consequently it requires some minor revisions.
Specific comments:
Line 215 The study selection should be better clarified with the description of software.
Figure 3 Please complete information for all articles as regard risk of bias.
Line 820 Authors should explain the potential mechanism for the increased risk in heart failure
At the end of the Discussion section authors should include limitations of this study.
Comments on the Quality of English LanguageMinor editing of English language is required
Author Response
Comment 1: The Manuscript (ID: brainsci-3087225) entitled “Non-ergot dopamine agonists and the risk of heart failure and other adverse cardiovascular reactions in Parkinson's disease” evaluated an interesting topic on the treatment of Parkinson’s disease. The aims of this review were to evaluate the risk of heart failure and other adverse cardiovascular reactions in PD patients who received a non-ergot dopamine agonist, compared with other anti-PD pharmacological treatments, placebo, or no intervention. Results indicated increased risk of heart failure in patients treated with pramipexole compared to other interventions. In general, the Manuscript is well written and clear to understand, consequently it requires some minor revisions.
Response 1: We thank the Reviewer for thoroughly reviewing our systematic review and for providing constructive feedback on how we may improve our manuscript. Furthermore, we are pleased that the Reviewer found our study topic interesting and an important contribution to the literature on the pharmacological management of Parkinson’s disease, and that they believe our manuscript is well written and clearly presented. Per the Reviewer’s recommendations, we have made the requested minor revisions to our manuscript.
Comment 2: Line 215 The study selection should be better clarified with the description of software.
Response 2: Per the Reviewer’s recommendation, we have provided information about DistillerSR, the review software used in our study, within the referenced section of our study methods. This includes details about important DistillerSR features that were leveraged to complete our systematic review, such as duplicate reference detection, multiple reviewer screening, conflict resolution, documentation of exclusion reasons, risk of bias and quality assessments, and data extraction via custom forms.
Comment 3: Figure 3 Please complete information for all articles as regard risk of bias.
Response 3: We thank the Reviewer for highlighting studies without risk of bias ratings in Figure 3. Studies within this figure that do not have a risk of bias rating for any of the examined domains represent the 8 non-randomized studies (4 case-control & 4 cohort) that were included in our review. It is common practice for systematic reviews to include all included studies in the risk of bias summary figure by study, including non-randomized studies. Notwithstanding, we note that the original caption describing this figure did not include details about the non-randomized studies without any risk of bias rating. Accordingly, we have revised the figure caption to include a description that studies without ratings for all domains represent non-randomized studies. We are happy to further revise the figure or caption as necessary.
Comment 4: Line 820 Authors should explain the potential mechanism for the increased risk in heart failure.
Response 4: The Reviewer raises an excellent point regarding the potential mechanism for the observed increased risk of heart failure with pramipexole, but not with ropinirole. Although the potential mechanism remains a mystery, we have added the following text to the referenced section of our discussion to offer a hypothesis as to the underlying mechanism contributing to heart failure following pramipexole use in Parkinson’s disease.
Dopamine agonists reduce peripheral resistance, increase salt and water excretion, reduce endothelial activity, and mitigate insulin resistance by activating dopaminergic and non-dopaminergic receptors. However, only cabergoline and pramipexole have been connected with HF, the former by producing heart valve fibrosis after activation of 5-HT receptors. The potential mechanism for the increased risk of heart failure with pramipexole remains a mystery. The cardiovascular effects of dopamine agonists cannot fully account for the increased risk in heart failure, as ropinirole has the same effects and yet does not increase heart failure. Pramipexole, but not ropinirole, is excreted unmetabolized through the kidney. It can be hypothesized that it might alter the intra-renal renin-angiotensin system, which could produce renal inflammation and liquid retention, which might contribute to heart failure.
Comment 5: At the end of the Discussion section authors should include limitations of this study.
Response 5: We have structured our manuscript’s discussion according to best practices and essential headings described in the Cochrane Handbook for Systematic Reviews of Interventions. We have therefore included the following paragraph that describes select study limitations within the Potential biases in the review process section of our discussion:
We identified few limitations to our review process. Due to many interventions and outcomes being examined in this study, we included specific adverse cardiovascular outcomes and employed validated harm filters in our electronic searches of the literature. As with other harm reviews, the search results may have omitted some studies that reported harms as a secondary outcome. It is also possible that some relevant studies retrieved by our searches were unintentionally excluded as a result of the inadequate reporting of all outcomes and adverse events. Therefore, it is plausible that some of our reported associations may be subject to bias due to selective outcome reporting or missing data. The effects of this bias were mitigated by our supplementary hand searches of trial registries and conference proceeding databases. Moreover, JAGC and YF reviewed the full text of efficacy studies that were believed (based on the abstract) to have also examined adverse events.
The aforementioned paragraph describing our study limitations is in the latter portion of our discussion; it precedes our overall conclusions. Accordingly, we propose maintaining the current placement of this paragraph in our discussion, though are happy to make any further edits to the description of our study limitation and discussion at the request of the Editor.
Comment 6: Minor editing of English language is required.
Response 6: We have thoroughly reviewed our manuscript for typos and correct grammar. Minor edits have been made to certain manuscript sections in order to ensure that the language used is clear, concise, and error-free.
Reviewer 3 Report
Comments and Suggestions for Authors
Parkinson disease (PD) is caused by neurodegeneration of dopaminergic neurons from pars compacta of substantia nigra from brainstem.
In symptomatic PD levodopa is gold standard, but also, dopamine agonists (DA).
DA used in PD treatment are of 2 types: ergot- derivates (cabergoline, bromocriptine, lisurid, pergolide) and non-ergot derivates (pramipexol, ropinirole, rotigotine, apomorphine, piribedil).
The ergot-derivates DA use in PD patients was associated with cardiovascular side effects: orthostatic hypotension, valvopathy, heart failure. Pergolide was withdrawn from US due to valvular heart damage.
New evidences suggest that also non-ergot derivates DA used in PD patient could be associated with cardiovascular adverse events.
The authors evaluated 36 randomized clinical trials (RCT) and 8 non-randomized studies (NRS) in order to evaluate the risk of developing heart failure and other cardio-vascular events in PD patients receiving non-ergot DA versus other PD treatments or placebo. One RCT found no significant difference between ropinirole (non-ergot DA) and bromocriptine (ergot DA) regarding cardiovascular risk. Also, 3 case-control studies reported risk of hear failure with non-ergot DA treatment in PD patients.
Patients with pramipexole had an in creased risk of peripheral oedema compared to placebo. On the other hand, patients with rotigotine had a lower risk of orthostatic hypotension, compared to placebo.
This is a verry well documented review regarding the cardiovascular adverse event of non-ergot dopamine agonists in Parkinson Disease treatment.
Author Response
Comment 1: Parkinson disease (PD) is caused by neurodegeneration of dopaminergic neurons from pars compacta of substantia nigra from brainstem.
In symptomatic PD levodopa is gold standard, but also, dopamine agonists (DA).
DA used in PD treatment are of 2 types: ergot- derivates (cabergoline, bromocriptine, lisurid, pergolide) and non-ergot derivates (pramipexol, ropinirole, rotigotine, apomorphine, piribedil).
The ergot-derivates DA use in PD patients was associated with cardiovascular side effects: orthostatic hypotension, valvopathy, heart failure. Pergolide was withdrawn from US due to valvular heart damage.
New evidence suggests that also non-ergot derivates DA used in PD patient could be associated with cardiovascular adverse events.
The authors evaluated 36 randomized clinical trials (RCT) and 8 non-randomized studies (NRS) in order to evaluate the risk of developing heart failure and other cardio-vascular events in PD patients receiving non-ergot DA versus other PD treatments or placebo. One RCT found no significant difference between ropinirole (non-ergot DA) and bromocriptine (ergot DA) regarding cardiovascular risk. Also, 3 case-control studies reported risk of hear failure with non-ergot DA treatment in PD patients.
Patients with pramipexole had an in creased risk of peripheral oedema compared to placebo. On the other hand, patients with rotigotine had a lower risk of orthostatic hypotension, compared to placebo.
This is a very well documented review regarding the cardiovascular adverse event of non-ergot dopamine agonists in Parkinson Disease treatment.
Response 1: We thank the Reviewer for taking the time to thoroughly review and provide feedback on our systematic review that evaluated the risk of heart failure and other adverse cardiovascular reactions among patients with Parkinson’s disease treated with non-ergot dopamine agonists. Moreover, we appreciate the Reviewer’s recognition that our review included both randomized and non-randomized sources of evidence, and that our review serves as a comprehensive resource on the topic of adverse cardiovascular reactions to non-ergot dopamine agonists in the treatment of Parkinson’s disease.
Reviewer 4 Report
Comments and Suggestions for Authors
The objectives of this review were to evaluate the risk of heart failure and other adverse cardiovascular reactions in PD patients who received a non-ergot DA, compared with other anti-PD pharmacological interventions, placebo, or no intervention. The idea of this study - is interesting; this review is gut structured and well-written, unfortunately, this manuscript needs some improvements and corrections before publishing may be possible.
General points:
Please add a list of abbreviations before References section to your manuscript.
Special points:
Figures
Please provide a better quality Figures: Figure 1, Figures 4-29.
Tables
Table 1. Characteristics of included studies: please move this Table to the Supplement.
Author Response
Comment 1: The objectives of this review were to evaluate the risk of heart failure and other adverse cardiovascular reactions in PD patients who received a non-ergot DA, compared with other anti-PD pharmacological interventions, placebo, or no intervention. The idea of this study - is interesting; this review is gut structured and well-written, unfortunately, this manuscript needs some improvements and corrections before publishing may be possible.
Response 1: We thank the Reviewer for thoroughly reviewing and providing detailed feedback on our systematic review. Furthermore, we thank the Reviewer for their recommendations on how we may improve the quality and readability of our manuscript. We agree with all of the Reviewer’s recommendations and have revised our manuscript accordingly.
Comment 2: Please add a list of abbreviations before References section to your manuscript.
Response 2: We agree with the Reviewer that the inclusion of a list of abbreviations would be helpful to readers. Per the Reviewer’s recommendation, we have included a list of abbreviations immediately prior to the list of references in our manuscript.
Comment 3: Please provide a better quality Figures: Figure 1, Figures 4-29.
Response 3: Per the Reviewer’s recommendation, we have included high-resolution versions of all manuscript figures in our revised submission.
Comment 4: Table 1. Characteristics of included studies: please move this Table to the Supplement.
Response 4: Per the Reviewer’s recommendation, we have moved the Characteristics of included studies table to the Supplementary Material (Supplementary Material 4), and have updated in-text references to all supplementary tables as appropriate.
Round 2
Reviewer 4 Report
Comments and Suggestions for Authors
Thank you for all corrections.